# Visualizing Superman: Artistic Strategizing in Early Representations of the Archetypal Man in Comic Books

**Bar Leshem** 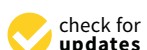

Department of the Arts, Faculty of Humanities and Social Sciences, Ben-Gurion University of the Negev, Be'er Sheva 8410501, Israel; barlesh@post.bgu.ac.il

**Abstract:** In 1933, Jerry Siegel and Joe Shuster, two Jewish teenagers from Ohio, fashioned an ideal personality called Superman and a narrative of his marvelous deeds. Little did they suspect that several years after conceptualizing the figure and their many vain attempts to sell the story to various comic book publishers, their creation would give rise to the iconic genre of comic book superheroes. There is no doubt that the Superman character and the accompanying narrative led to Siegel and Shuster, the writer and artist, respectively, becoming famous. However, was it only the appealing character and compelling narrative that accounted for the story's enormous popularity, which turned its creators into such a celebrated pair, or did the visual design play a major part in that phenomenal success? Recent years have seen a burgeoning interest in the comic book medium in several disciplines, including history, philosophy, and literature. However, little has been written about its visual aspect, and comic book art has not yet been accorded much recognition among art historians. Since the integration of storyline and art is what allow the comic book medium to be unique and interesting, I contend that there should be a focus on the art as well as on the narrative of works in comic books. In the present study, I explore the significance of the visual image in the prototype of the Superman figure that Siegel and Schuster sold to DC Comics and its first appearance in the series *American Comic Books*. I argue that although the popularity of Superman's first appearance was due to the conceptual ideals that the character embodied, the visual design of the ideal man was also an essential factor in its success. Accordingly, through a discussion of the first published Superman storyline, I emphasize the artistic-visual value of the figure of this protagonist in particular and the comic book medium in general.

**Keywords:** comics studies; comics art; Superman; mass media; pop culture

## 1. Introduction

In 1933, Jerry Siegel and Joe Shuster, two Jewish teenagers from Ohio, fashioned an ideal personality they called Superman and developed a narrative about his marvelous deeds. Little did they suspect that several years after conceptualizing the figure and their many vain attempts to sell the story to various comic book publishers and syndicates, their creation would give rise to the iconic genre of comic book superheroes.[1] There is no doubt that the Superman character and the accompanying storylines led to Siegel and Shuster, the writer and artist, respectively, becoming famous. However, was it only the appealing character and compelling narrative that accounted for the comics' enormous popularity, which turned its creators into such a celebrated pair or did the visual design account for a major part in that phenomenal success?

In the present study, I explore the significance of the visual image in the prototype of the Superman figure that Siegel and Shuster sold to Detective Comics Publishing, commonly known today as DC and its first appearance in the comic book series called *Action Comics*. I also discuss the emergence of the Superman iconography as a case study in artistic strategizing in the marketing of works of art in the early twentieth century. One of the main reasons for the popularity of Superman's first appearance as a hero was due to

the conceptual ideals that the character embodied, as was conveyed by various scholars. Nevertheless, I argue that the visual design of the ideal man, as well as the appealing visual aspect of the narrative, was also an essential factor in its success. Accordingly, through a discussion of the first few published Superman storylines, I emphasize the artistic-visual value of the figure of this protagonist and the illustrated narrative.

In a 1983 interview for *Nemo: The Classic Comics Library* magazine, Siegel and Shuster recalled the formalizing of the Superman figure back in the 1930s (Andrae et al. 1986, pp. 6–19). It seems that even in the early days of the Superman epoch, the two young men sought to create not only a comic book hero but a true pop culture phenomenon. Among the many pop culture heroes that inspired the creators of the first superhero, some of which are discussed further on, Siegel noted the pulp fiction protagonist Tarzan and the marketing strategies that his creators and producer had constructed.[2] This, Siegel recalled, affected his and Shuster's approach for their own creation:

"One day, I read an article in some leading magazine of the time about how Tarzan was merchandised by Stephen Slesinger so successfully. And I thought: Wow! Superman is even more super than Tarzan: the same thing could happen with Superman. And I mentioned it to Joe [Shuster] . . . and he had made a big drawing of Superman showing how the character could be merchandised on boxtops, T-shirts, and everything. We put this merchandising business into one of the very early Superman stories. The publisher looked at it and thought it was a good idea, and Superman had been a terrific earner from character merchandising ever since" (Andrae et al. 1986, p. 15).

Shuster then added: "In this drawing we just let our imagination run wild. We visualized Superman toys, games, and a radio show . . . and Superman movies. We even visualized Superman billboards" (ibid).[3]

The thoughts of those two young men at the beginning of their careers suggest their aspirations and their vision regarding their comic book protagonist. The financial success might not have been their priority, but clearly their eagerness to bring this character to life, not only through the comic book pages but also through every young person's appurtenances was apparent from the beginning.

The visual side of the Superman design scheme is clear from the quote above as a starting point for developing a marketing strategy for the comics. This is fertile ground for the present discourse: the marketing strategies for Superman as a figure and as the protagonist of a narrative. When examining the subject of artistic strategizing in the marketing of works of art, there are several possible approaches. For example, one might be an analysis of the artist-patron relationship. Here, determining the reasons a patron commissioned an artwork from a specific artist is relevant or, alternatively, the artist's strategies for attracting possible patronage or, perhaps a specific patron.

Daniel Unger's study of Caravaggio's *Boy Bitten by a Lizard* as a self-advertisement strategy reveals an instance where an artist designed a work in order to market himself, using visual trends and features that were common in contemporary Rome (Unger 2016, pp. 21–50). Another approach in connection with artistic marketing is a consideration of the artist-audience relationship, which is especially relevant where the medium is compatible with the ideas of mass and/or pop culture. This involves the reception of an artwork or a notion by the audience, which becomes a sort of "modern patron" by consuming such objects (Wilson 1983, pp. 39–64; Gordon 1995, pp. 49–66).[4]

I contend that comic book marketing strategies employ an integration of these two approaches, as the medium reflects a combination of text and image, which could be interpreted, on the one hand, as art and, on the other, as a product of pop culture, which was, and still is, directed toward the masses. Thus, it gives rise to a challenging, interesting, and unique case study, which I discuss in the pages that follow.

As noted above, in this essay, I explore the success of the Superman comics through the early issues of *Action Comics*. To do so, I consider several visual aspects of the creation: the design of the figure of Superman and other archetypal characters, such as the villain of those earlier narratives and the female love interest; the concept of an illustrated narrative,

with its three main visual features, which helped the storyline progress, that is, emotions, humor, and action; and the closing panels, which were hybrids of a sort that related to the narrative but were also advertisements that appealed to the audience. Clearly, all of these aspects were designed to engage the audience with the character of Superman and the accompanying narratives, and thus enhance the sales of the comic books and Superman's rise to popularity. The novelty of the present study lies in its unique perspective: understanding the marketing strategies through analyses of the figures, the narrative, and the embedded advertisements.

After a brief review of the way Superman was created and an introductory discussion about Siegel and Shuster, I explore several contemporary trends that promoted and inspired the visual design of Superman. I present several methodological approaches that have been developed over the years in connection with the visual aspect of comics, specifically in comic books. I discuss the design of the Superman and other characters, the illustrated narrative, and the closing panels and suggest how they were incorporated into the comic books and their impact, deliberate or unwitting, on the marketing of the Superman comics.

## 2. The Visual-Verbal Narrative Art

Recent years have seen a burgeoning interest in the comic book medium among scholars in several disciplines, including history, philosophy, and literature. However, comic book art has not yet been accorded much recognition among art historians. In 1985, Will Eisner, an American writer and cartoonist, who was one of the earliest artists to work in the American comic book industry, published a book on sequential art (Eisner 1985). In the chapter devoted to imagery, he contends that the visual aspect of sequential art is a form of communication and that "the success or failure of this method of communicating depends upon the ease with which the reader recognizes the meaning and emotional impact of the image" (ibid., p. 14). In this paper, I discuss the visual aspects of the earlier issues of Superman from the perspective of the way in which they were meant to engage the audience in the storyline, constructing, as Eisner suggests, a form of communication between the creators and the readers/viewers.

As was implied in the previous section, the art-historical point of view plays a key role in analyzing the various illustrations, character design, and comics panels presented here. Throughout this paper, I discuss Shuster's artistic style and the way he portrayed emotions and movements, while using a minimal number of lines. A key term here is 'iconography,' which is used on several occasions. For example, when comparing the visual design of various figures and scenes, I look at the way other visual media inspired Shuster's creation. In the concluding section, I contend that Superman's visual design established the prevalent iconography of the comics male superhero and that the creators of many of the subsequent comics superheroes used Superman's design as a point of reference. In discussing the character's design, I employ the art historical method of analyzing other earlier and contemporary artistic traditions. This was discussed briefly in the previous section in connection with Unger's approach to Caravaggio's *Boy Bitten by Lizard*. Even though many of the visual elements that inspired Shuster's creation were not 'artistic' per se, since the comics medium itself can be regarded as both art and pop culture, other media forms can be thought of as visual-artistic sources of inspiration. All this is to say that this paper takes terms and approaches widely used in the art history field to analyze the first few issues of the Superman comics in the *Action Comics* series.

Before broaching other research published in this context, I note here some of the terms that I use throughout this paper. First, the concept defined as 'illustrated narrative' is derived from Robert C. Harvey's definition, which is presented in the next paragraph. I use it principally to describe the way Shuster portrayed the ideas in the storylines and scenes, that is, when I deal with the design or creation of a scene or a narrative continuity. 'Visual design' is a broader term, which I use in describing the design of characters (Superman, Clark Kent, Lois Lane, and the occasional villain), but also more broadly in discussions of the specific visual elements in the comics storylines. This is similar to the use of the

term 'illustrated narratives,' since, when discussing the way Shuster created the narrative's continuity and panel scenes, I also relate to the visual elements and designs that helped him develop them. Generally speaking, concepts such as 'visual design,' 'illustrations,' and 'illustrated narrative' might overlap, depending on the context.

In 1996, Robert C. Harvey, an American author, critic, and cartoonist published a book entitled *The Art of the Comic book: An Aesthetic History* (Harvey 1996). Comics (i.e., comic strips and later comic books) have been known as a defined medium from the late nineteenth century. Many scholars regard earlier artworks as the forerunners of this kind of medium, pointing to William Hogarth's (1697–1764) *A Harlot's Progress* paintings (1731–32) as the beginning of pictographic sequential art (Bartual 2010, pp. 83–105), but Harvey was among the first to have introduced a more academic jargon for the visual side of this discourse.

Harvey contends that a new vocabulary is needed when addressing this medium and that the language used to describe and analyze the medium has been based mainly on the cinematic or the literary scholarly field. He notes that the uniqueness of the comic book medium lies in the combination of the narrative and visual elements. "Good" comics (if they can be referred to terms of good and bad) are those in which there is no hierarchy between those two elements: the illustrations are supported by the text and vice versa and without either of those elements, the meaning would be lost. To refine the appropriate research jargon for comics, Harvey developed a new vocabulary that supports the combination of the art, the literature, and the cinematic disciplines. Among his definitions are the narrative breakdown: "the division of a story into panel units"; composition: "the arrangement of pictorial elements within a panel"; layout: "the arrangement of panels on a page and their relative size and shape"; and style: "the highly individual way an artist handles pen or brush . . . " (Harvey 1996, p. 9). Visually, he relates to the "visual significance of the narrative" and stresses that when examining comics, we must pay attention to the relationship between the storyline, that is, the literary narrative, and the way in which it is portrayed through the visual element of illustrations and the way the panels are divided (ibid., pp. 8–10). Although the jargon itself seems common to the abovementioned disciplines, in this paper, I focus on the art historical approach.

A more recent study of visuality in comics was published in an edited volume entitled *Comic Studies: A Guidebook* (Hatfield and Beaty 2020). Chapter 13, "Words and Images," by Jan Baetens develops earlier ideas concerning the historical study of the relationship between words and images, suggesting that the text has a linear role whereas the illustrations have a more spatial aspect (Baetens 2020, pp. 193–209). He also contends that eliminating the hierarchy between the visual and the verbal means "that images can no longer be a mere supplement to an underlying story" (ibid, p. 195).

Having concluded that the uniqueness of the medium lies, for most part, in the verbal–visual relationship and that there must be some sort of reciprocity between the two, I now turn to a case study of the Superman narrative. It is important to keep in mind Eisner's note on the relationship between imagery and audience, as I contend that the enormous success of the Superman phenomenon lies in this specific relationship. We should keep the points raised here in mind when we consider the visual-artistic strategies for selling the Superman comic narrative and its rise to popularity.

## 3. The Reign of Superman

Superman, the first superhero in that genre of comic books hardly needs any introduction, so here I simply offer a brief review of the historical development of the Superman comics. Superman was the creation of two Jewish boys from immigrant families living in Ohio. Jerry Siegel, a nerdy teenager who spent most of his time daydreaming about science fiction and writing for his high-school's newspaper, met Joe Shuster, a shy slender boy with no self-confidence, in school. Owing to their love of pulp fiction, Douglas Fairbanks' movies, and comic strips, the two quickly became friends and partners. Siegel appreciated Shuster's artistic talent and the two boys started working on comic storylines, mostly in the form of comic strips.[5]

In 1932, the two boys came up with the idea of a comic narrative that they called "The Reign of the Superman." Their first Superman was a bald-headed villain, who used his superhuman powers for his own gains.[6] A few months after creating this storyline, Shuster began to develop a new character with the same name, who was a hero rather than a villain. When asked, in the *Nemo* interview, why they changed this character, Siegel responded: "Obviously, having him a hero would be infinitely more commercial than having him a villain. . . . Creating a successful comic strip with a character you'll hope will continue for many years, it would definitely be going in the wrong direction to make him a villain" (Andrae et al. 1986, p. 10). Even though Siegel and Shuster's intention was to make him more sellable, the Superman storyline, with its new and improved vision of a hero, created in early 1933, did not sell until 1938, when Detective Comics needed new material for their new comic book series *Action Comics*. The rest of the history of the Superman comics is well known: Detective Comics ordered thirteen pages from Siegel and Shuster for the first issue, for $10 a page, with the boys waiving their rights. With increasing sales each month, by the time of the fourth issue, youngsters were running to the newspaper stands to ask for the comics "with Superman in it" (Harvey 1996, pp. 16–19; Gordon 2017, pp. 4–8).

Superman's rise in popularity coincided with an interesting time in the United States. America and the world were getting used to a new reality after the Great Depression and the country's leaders were promoting the notion of the "American Dream" (Filc 2017, pp. 47–60). The idea of an "American way of life" had developed in the 1930s and it seems that mass and pop culture fostered the concept among the public (Eaton 2013, pp. 28–39; Chambliss and Svitavsky 2013, pp. 6–27). As he was introduced on the first page of the Superman story in *Action Comics* #1, Superman was the "Champion of the oppressed, the physical marvel who had sworn to devote his existence to helping those in need!" which seemed to fit perfectly with that notion (Figure 1).[7]

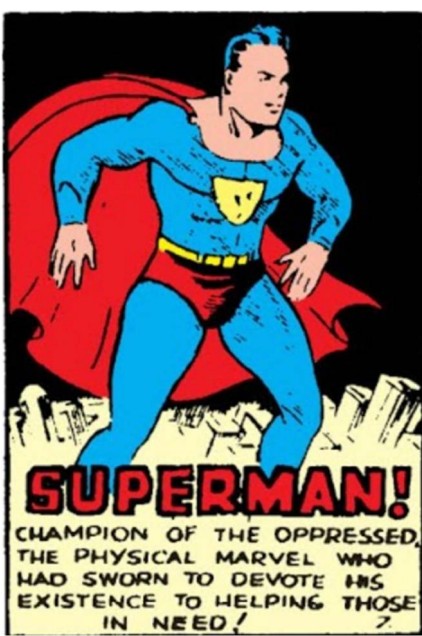

**Figure 1.** Joe Shuster, "Superman! Champion of the Oppressed" (panel No. 7), 1938. Siegel and Shuster (June 1938), "Superman", *Action Comics* #1 (Siegel and Shuster 1938a, p. 8). © DC Comics.

In Superman's early appearances in *Action Comics*, he did not fight supervillains or mad scientists and he did not have an arch-nemesis supervillain such as Lex Luthor. He fought the injustices that were evident in the everyday life of middle-class America: in *Action Comics* #1, he sought to acquit a woman and succeeded a few minutes before she was wrongly to be executed and confronted a wife-beater and a corrupt politician. In succeeding issues, each storyline was devoted to one principal narrative: in the second issue, Superman

dealt with a man who tried to stir up a war and in the third he rescued a coal miner trapped by a cave-in.[8] In those first issues, he did not battle a "higher power" that threatened his city or the United States but rather dealt with ordinary everyday matters.

Superman's alter ego was a shy journalist, who was constantly being rejected by his beautiful co-worker, Lois Lane. He wore glasses and the highlight of his day was to photograph Superman in action. As has been noted in several studies, and by Siegel and Shuster themselves, Clark Kent was a fictional version of the creators themselves, who, like Kent, did not have any luck with women, were shy, and in the early stages of their lives also dreamed of a job in journalism (Chambliss and Svitavsky 2013, pp. 19–20). When asked about the conception of Superman's dual identity, Siegel recalled:

> Clark Kent grew not only out of my private life, but also out of Joe's [Shuster]. As a high school student, I thought that someday I might become a reporter, and I had crushes on several attractive girls who either didn't know I existed or didn't care I existed. . . . It occurred to me: What if I was real terrific? What if I had something special going for me, like jumping over buildings or throwing cars around or something like that? . . . The concept came to me that Superman could have a dual identity, and that in one of his identities he could be meek and mild, as I was, and wear glasses, the way I do. . . . By coincidence, Joe was a carbon copy [of me]. (Andrae et al. 1986, p. 11)

Superman's involvement in everyday American life and Kent's identifiable nature for many of the youngsters who became enthusiastic readers of the superhero genre contributed to the character's popularity.

After the enormous success of the *Action Comics'* Superman storylines, in 1939, DC Comics launched a new comic book series devoted entirely to Superman, which was soon followed by a radio show and later on cartoons, films, and TV series. In 1940, a 75-ft Superman balloon was included in Macy's Thanksgiving Day Parade, clear proof of the popularity and hype around the first superhero (Figure 2; Grippo and Hoskins 2004, p. 50). Apart from the Thanksgiving balloon, Superman was merchandised in Macy's store, which also dedicated a "Superman Day" at the World's Fair in July 1940. This was promoted by comics issues called "New York World's Fair Comics" featuring Superman.[9] These examples demonstrate the character's early rise in popularity. Further, in 1941, *The Saturday Evening Post* featured a major piece on Siegel and Shuster, focusing on the young creators' financial and creative success (Kobler 1941, pp. 14–78). Clearly, Siegel and Shuster's dreams of success and the merchandising, and franchising of Superman had come true. However, why did it come true? What was the difference between Superman and the other pop culture heroes, as Tarzan, Doc Savage and other pulp magazines protagonists of the time? Was it the comics format that differentiated between the heroes of the pulp fiction, for example, and the comics hero? Was it the Superman narratives? Or was it the successful creation of the archetypal ideal man and his embodiment in this unique medium? More specifically, what part did the visual side of these comics contribute to their success?

Some scholars have explored Superman's evolution as an icon over the years, noting that the early issues of *Action Comics* furnished the raw beginning of the superhero genre. Umberto Eco, for example, focused primarily on the 1960s Superman comics and on Superman's mythical roots, whereas Ian Gordon referenced his iconic qualities (Eco and Chilton 1972, pp. 14–22; Gordon 2017). Other scholars examined the rise of these comics from a biographical perspective, focusing on the Jewish roots of the creators, and how they were embedded in the figure of Superman and the character's configuration on the basis of Siegel and Shuster's own experiences and aspirations (Harvey 1996, p. 21; Daniels 1998; Andrae and Gordon 2010; Brod 2012; Lund 2016).

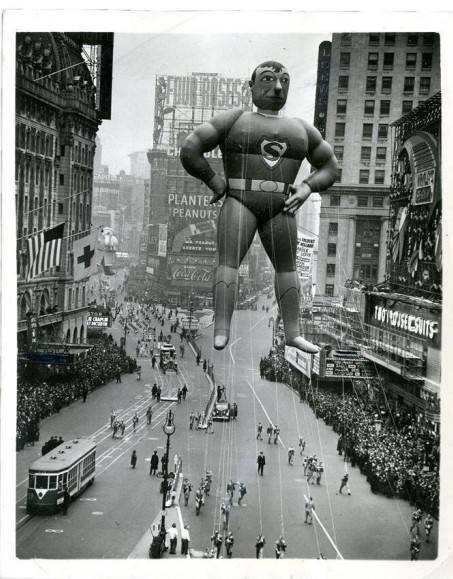

**Figure 2.** Tony Sarg (designer), *Superman Balloon in the Macy's Thanksgiving Parade*, 1940, 2286 × 1341 cm. Photo: Wiki Fandom, Creative Commons.

Visually, Shuster's illustrations, mainly from the earlier years, were subjected to some criticism. Harvey mentioned that some have considered his illustrations "primitive" (Harvey 1996, p. 19), whereas Jules Feiffer noted that, "In drawing style, both in figure and costume, Superman was a simplified parody of Flash Gordon." However, Feiffer does seem to give Shuster some credit by adding that, "Shuster represented the best of old-style comic book drawing. His work was direct, unprettied—crude and vigorous; as easy to read as a diagram" (Feiffer 1965, pp. 19–20).

The coloring of the comics in the early issues was rather basic, with pure local colors decorating the background of each panel, each with a different background shade, and the characters themselves were colored the same way. The coloring technique of this printing was the CMYK color model, which featured cyan (a shade between green and blue), magenta (a shade between purple and pink), yellow, and key (black), which comprised the palette for the basic printing technique of the time (Palacio and Vit 2009, p. 59; Legion of Andy n.d., Section 6). In terms of influences, Shuster often disclosed his admiration and the inspiration he took from the pulp magazines of the science fiction genre illustrator Frank R. Paul, well known for his cover art and interior illustrations. Paul was often regarded as the originator of science fiction art, and his covers evolved into the dominant style of this genre and medium in the 1930s (Mitchell 1984, pp. 121–32).

Harvey and Feiffer define Shuster's style as somewhat raw. Indeed, although Shuster sought for realism in his artwork and tried to avoid the more 'cartoony' style that was generally seen in comic strips, his illustrations were simple yet affective (Andrae et al. 1986, p. 12). With a few lines to construct the characters' faces, his minimalist line drawings pictured distinctive facial features, facial expressions, and emotions, as well as movement—all of which are discussed in the following pages.

## 4. The Visualization of Superman and Other Archetypal Characters

The first important aspect that should be addressed when discussing Superman's visuality is the design of the protagonist himself. From a literary or even ideological point of view, Superman is portrayed as the ideal man, who fights for a better society. As many researchers emphasize, his idealism traces back to the image of the hero in history, principally from the classical and biblical worlds, and he is often thought of as a modern-day mythical hero (Eco and Chilton 1972, pp. 14–22; Caruth 1968, pp. 1–12). As early as in 1940, for example, Slater Brown wrote in *The New Republic* that Superman was "handsome

as Apollo, strong as Hercules, chivalrous as Lancelot, swift as Hermes. . . . A Hero God" (Brown 1940, p. 301).

An interesting image from the first Superman comics narrative supports and expands on Brown's statement by comparing Superman to the Greco-Roman hero Herakles, not only conceptually but also in a more visually concrete way: the first *Action Comics'* Superman storyline told of the hero's origins. In the first row on the first page, the infant Superman is lifting an armchair above his head, and a man and a woman are stunned by the sight (Figure 3).[10] Although there is no evidence that Shuster was familiar with old representations of the ancient hero, there are some similarities, both in the narrative and the visual aspects, to the ancient myth of the infant Herakles holding off a snake (Figure 4). The idea of an infant possessing superhuman strength is widely known in ancient history and art through the Herakles myth, and this was translated, even if not directly, to the Superman narrative and illustrations.[11] Superman continued to be shaped visually like that ideal hero throughout the comics' history.[12]

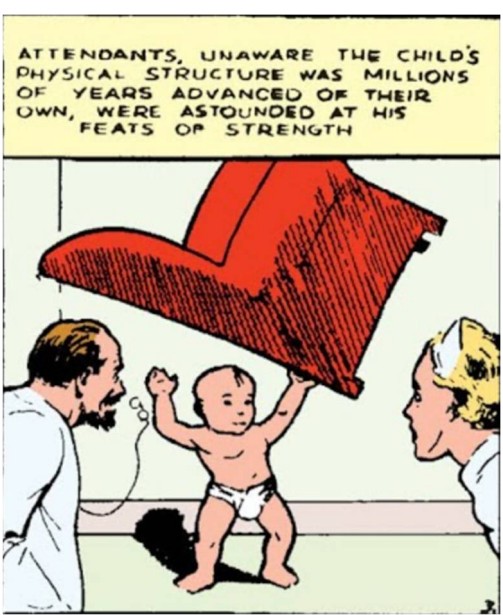

**Figure 3.** Joe Shuster, Infant Superman (panel No. 3), 1938. Siegel and Shuster (June 1938), "Superman", *Action Comics* #1 (Siegel and Shuster 1938a, p. 8). © DC Comics.

Superman and Clark Kent represent two ways of addressing an audience: Superman is the superhero that the American youth aspired to imitate and Clark Kent, the shy, awkward journalist who tries to hide his secret identity, is a character with which many of these same young people can identify. Visually, except for their clothes Kent and Superman are not all that different. Superman's iconic costume is the blue bodysuit with red trunks,[13] a red cape, and a yellow symbol on his chest with the red letter S. His facial features are similar to those of Hollywood movie stars at the time, such as Carry Grant and Clark Gabel, who were mentioned as inspirations by the creators themselves. In the *Nemo* interview, for example, one of them noted that Superman was "conceived as being like the ideal Hollywood romantic hero of the time" (Andrae et al. 1986, p. 12).

In the 1930s, Gable and Grant were the embodiments of tough, ideal men, who, unlike earlier film protagonists, were neither rich nor violent.[14] They played down-to-earth, serious guys, who were admired by men and loved by women. Moreover, both of them visually represented the ideal man of that era, with their black silky hair, dimples, and charm. Most of those characteristics are also featured in Superman himself, who was and is the embodiment of the pop-culture hero. It is not surprising that the young Siegel and Shuster, who often went to the movies, designed their own hero in the archetypal character of the ideal cinematic man of the time.

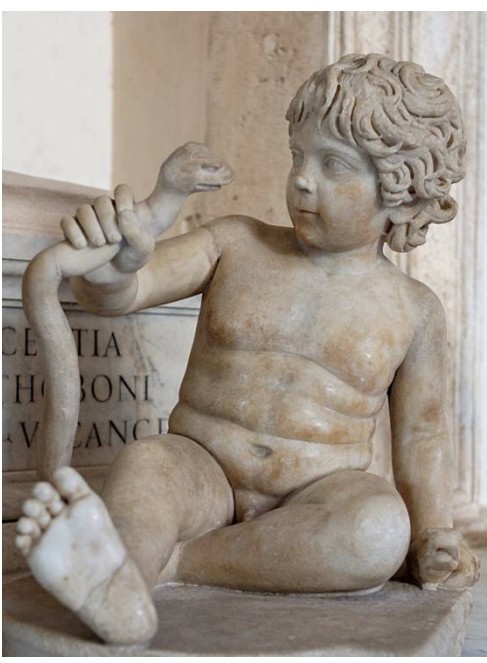

**Figure 4.** *Herakles as a Boy Strangling a Snake*, 2nd Century CE, Marble, Rome: Musei Capitolini. Photo: Wikipedia Commons, Public Domain.

Another prominent figure in the design of Superman was the 1920s silent-film star, Douglas Fairbanks Sr., who was a charming character in a different way than Gable or Grant. Superman's creators often indicated that they loved Fairbanks' films and much of the early years' Superman storylines were inspired by such movies such as *Robin Hood* and *The Black Pirate*. Superman's stance—legs spread apart and hands on his hips—was taken directly from Fairbanks's screen image (Figure 5a,b).[15]

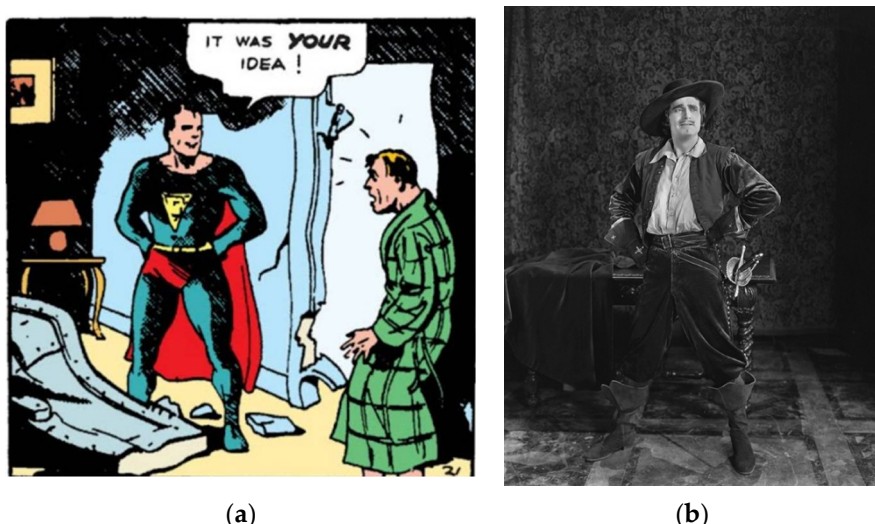

(**a**)                      (**b**)

**Figure 5.** (**a**) Joe Shuster, Superman in a Fairbanks Stance (panel No. 21), 1938. Siegel and Shuster (June 1938), "Superman," *Action Comics* #1 (Siegel and Shuster 1938a, p. 10). © DC Comics; (**b**) *Douglas Fairbanks, The Iron Mask*, 1929. Photo: Wikipedia Commons, Public Domain.

It seems that the young creators, visualizing the ideal man, and enthusiastic movie fans managed to combine recent popular fashions with the ideal heroes of previous decades. The result was a masculine, tough man, with a strong square chin; the occasional quirky smile, which reflected Fairbanks' film characters' humorous view on life; dark hair with a

single frontal curl, which copied Gable's hairstyle at that time; and a heroic stance taken directly from images in silent movies (Figure 6).

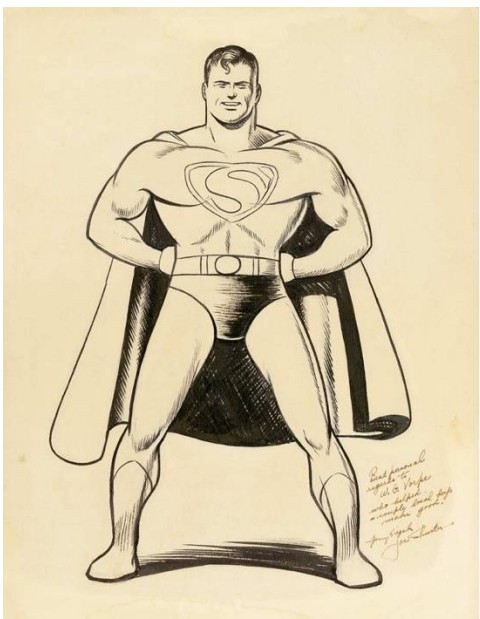

**Figure 6.** Joe Shuster, Superman, ink drawing, probably 1970s. © DC Comics.

Along with the Hollywood influence, other media influenced the shaping of Superman's image. The most prevalent aspect in the research field is the "Strongman" figure Siegmund Brietbart and the bodybuilding magazines (Andrae and Gordon 2010, pp. 55–61). It is well known that, owing to his shyness, from a young age Shuster started to build up his body to increase his self-confidence. He was often to be found in the school's gym and continued to work on his fitness throughout his adult life, a fact that was noted, for example, in an issue of *Saturday Evening Post* in 1941. Addressing their new financial success, the journalist John Kobler wrote: "To Joe Shuster the most important thing money has bought is the chance to build up his body. For years he tried Lionel Strongfort correspondence school methods. . . . Now he lifts weights three times a week in Barney Kofron's gym" (Kobler 1941, p. 76).

Clearly, Superman's strong physique was also inspired by Shuster's own interests. It is also reasonable to assume that many young readers were familiar with the Strongman figure and those magazines. One can look, for example, at one of the most popular magazines of those days, a 1937 issue of *Physical Culture Magazine*. On the seventh page, a large advertisement reading "Lend Me Your Body for 7 Days" shows a masculine man in shorts, the famous bodybuilder Charles Atlas, who had his own technique for getting into shape. The text around the image gives the man's muscle measurements, demonstrating the desirable body type (Figure 7; Atlas 1937, p. 10). Atlas was not only well known from fitness magazines but was also featured in comics that dealt with the era's ideal of hypermasculinity (Andrae and Gordon 2010, p. 57). Another example is the 1930s physical culture *Superman* Magazine, which often featured nude male figures in heroic poses. This magazine not only portrayed the ideal male figure of the time on its covers but had the same name that was associated with the first comics superhero (Duncan et al. 2015, p. 195).

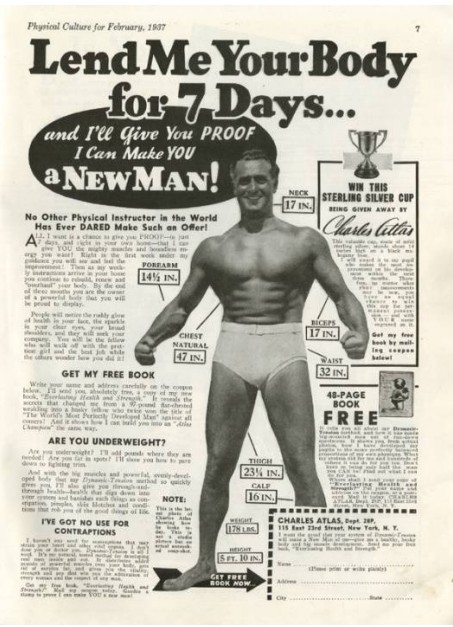

**Figure 7.** Charles Atlas, "Lend Me Your Body for 7 Days," Physical Culture vol. 77 (2), 1937, p. 10. https://dmr.bsu.edu/digital/collection/PhyCul/id/13386 (accessed on 27 August 2021). Source: Ball State University Archives and Special Collections.

As I noted earlier, the pulp magazine heroes of that time, mainly those of the science fiction genre, also had an influence on Siegel and Shuster. Both were avid readers of the genre and often voiced their admiration and acknowledged the impact that it had on their own creation. Siegel and Shuster thought about Flash Gordon, John Carter of Mars, Tarzan, and other such heroes when they were trying to re-imagine their superhero character from villain to hero. Similar to Hollywood's heroes at the time and the ideal men portrayed in fitness magazines, the pulp fiction protagonists were also often manly, clean-shaven man who fought evil and saved damsels in distress (De Silva 2004; Chambliss and Svitavsky 2008, pp. 1–33). A further prominent figure who inspired Superman's abilities was Hugo Danner, the protagonist of the 1930s novel *Gladiator*, written by Phillip Wyle. Superman's leaping abilities, superstrength, and speed, which were all featured in the first pages of *Action Comics* #1, were taken directly from the description of Danner in Wyle's novel (Duncan et al. 2015, p. 193; Wilcock 2016).

Looking at all of the elements that inspired the creation of the first comics superhero, one can see that many strategies were involved in the actions taken to address readers and familiarize them with this new character and genre. To shape the new figure of Superman, Shuster, as the illustrator, took inspiration from many of the genres that he knew and loved. Superman was not a character with whom his readers could identify, but someone they aspired to be like, or more specifically to this discussion, someone that they wished to look like.

Before broaching the subject of the illustrated narrative, it is important to briefly examine the design of other characters in the early Superman storylines: the archetypal villain and the female figure. From the very first issue, Lois Lane featured as Kent's and Superman's love interest. She was described as a hot-headed and sassy reporter, who disliked Kent and was in love with Superman. Her figure, like that of Superman, was derived largely from the movies, where, in the 1930s, the female news reporter was a popular cinematic trope, for example, Torchy Blane, played by Glenda Farrell (Daniels 1998, p. 20). Other figures seem to follow along this line of identifiable typical characters in films, books, and pulp magazines. The villains of the early Superman storylines were not the supervillains we know today, but as noted above, were much more mundane. Most of them had some characteristic that differentiated them from the "good guys": some had thin mustaches, some were fat, some had pointy chins, and some were imaged in suits that

stood out because of their colors or patterns. The villains were not ideal masculine figures, were often portrayed in a somewhat grotesque manner, and were sometimes evil looking.

The first issue of *Action Comics* featured the new Superman character on its cover. Although he was not on the covers of the next several issues, he did adorn *Action Comics* covers from time to time before the advent of the *Superman Comics* (together with *Action Comics*, which continues to be published with Superman's storylines to this day). Along with the creation of Superman's visual attributes, study of the illustrative process of the narrative itself can also assist in the understanding of Siegel and Shuster's efforts to promote and sell their creation.

## 5. Illustrating the Narrative

When asked about their creative process, Shuster replied: "Jerry picked up the technique of visualizing the story as a movie scenario and whenever he gave me a script, I would see it as a screenplay. That was the technique that Jerry used, and I just picked it up." Later in the same interview, he disclosed that, "I [Shuster] was caught up in Jerry's [Siegel] enthusiasm, and I started drawing as fast as I could use my pencil. My imagination just picked the concept right up from Jerry" (Andrae et al. 1986, p. 11). Siegel, in his turn, revealed his admiration for Shuster's art, saying: "When I saw the drawings that were emerging from his pencil, I almost flipped. I knew he had matured a great deal since he had done the first Superman [from the early 'villainesque' version in "The Reign of the Superman"], and I thought he was doing a great job on the new art" (ibid., p. 12).

It is clear that Siegel and Shuster had movie storylines in mind when they created the early Superman comics, which is also apparent, as noted above, in the design of the principal protagonist and other archetypal figures. Thus, although Harvey attempted to detach the cinematic jargon when referring to comics and comic art, but in the context of the early years, it is hard to talk about one without the other. Nevertheless, Harvey made a valid point when he expressed the need to develop a new jargon for comic art. The following pages are devoted to analyses of several narrative panels from the first few issues of *Action Comics* in order to demonstrate the various ways that Shuster visualized Siegel's vision: the portrayal of emotions, the action scenes, and the humor.

Along with the design of the figures themselves, another aspect stands out when discussing the visual side of the storylines, which Harvey defines as 'illustrated narratives' (Harvey 1996, p. 9). When Superman debuted in *Action Comics'* first issue, the comic book medium was in its infancy. Although Shuster was a meticulous artist who paid attention to every detail and tried to move away from the 'cartoony' style that characterized "The Reign of the Superman," his work in the first few issues was somewhat similar to the style and the kind of humor that characterized the comic strips that preceded it. Harvey contends that Shuster failed to utilize the potential of the new comic book medium and stuck to the familiar patterns of the comic strips. Thus, although the occasional vertical panel in a Superman comic book accentuated a situation, most of them were similar to those featured in the comic strips. Nevertheless, even if the initial comic books did not utilize the potential of this new medium to the full, the continuity of action and scenes suggests the successful attempt to move away from one-strip comics to a more elaborate narrative. Further, Shuster seemed to have mastered three important elements—emotions, humor, and action—which may have enhanced the audience's positive response to the new comics.

Shuster visualized Siegel's visionary 'movie' action scenes using a few 'tricks': Superman in motion was depicted by diagonal lines surrounding him, which indicated his movements, as well as an occasional 'close-up.' Oftentimes, when Superman was battling with a villain, a criminal of some sort, the latter was imaged with a look of horror on his face: eyes open wide, his mouth crooked, and sometimes holding his face in his hands, with small lines around his face to show his fright. Superman, in turn, often appeared with a contented smile when he was, as always, the victor. That smile was inspired by Fair-

banks, who often threw his head back in laughter when faced with a dangerous situation (Andrae et al. 1986, p. 14).

A few examples can serve to demonstrate these elements. First, one needs to look no further than the cover of the first issue of *Action Comics* to find many of these features (Figure 8). That cover pictures one of the storyline's most significant scenes—the beginning of Lois Lane's infatuation with Superman. Taken from the ninth page of the issue, it shows some thugs harassing Clark Kent and Lois Lane, who are out on a date. Kent sidesteps them and does not assist Lane, probably to avoid revealing his powers, but after Lane is forced into the car, Superman arrives and saves the day by lifting the car and throwing the thugs out. On this cover illustration, which is slightly different than the interior panel, probably to make the first cover of the new series more of a draw as a stand-alone picture, Superman is at the center of the composition, holding the car high up over his head, one of his legs extended forward, and his cape flared out behind him to indicate his movements. The top section of the opaque yellow background is decorated with diagonal black and brown lines, which mark the action. The fleeing thugs were portrayed in a humorous way. One, on the back plane, is shown running with his hands extended forward, while another is on his knees, with a look of horror on his face. The most conspicuous figure is in the foreground on the left side of the picture: his eyes are open wide, his crooked mouth indicates his fear, and he is holding his head in his hands; his red dotted tie is blowing in the wind, indicating that he is also escaping from Superman. This first cover shows Shuster's unique talent in combining action, movement, and comic exaggeration in a single panel. The influence of earlier comics art is seen mainly in the portrayed humor, as until the 1930s American comic strips generally featured humorous storylines and exaggerated bodily gestures and facial expressions.[16]

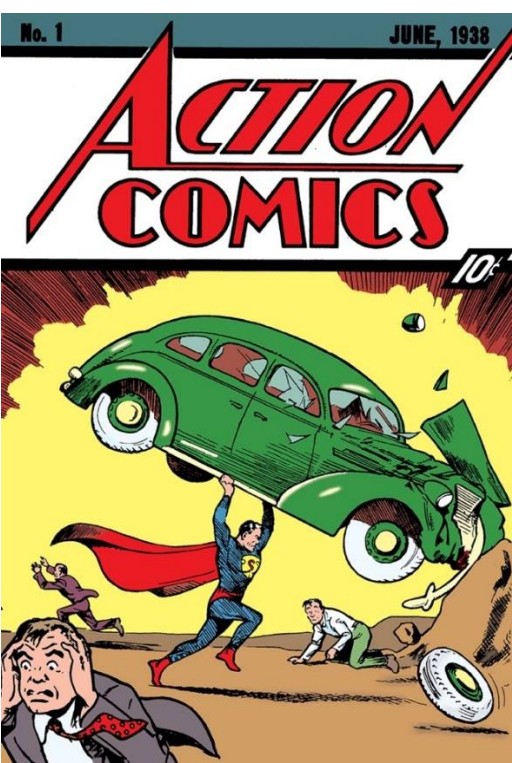

**Figure 8.** Joe Shuster, Cover of Action Comics #1. Siegel and Shuster (June 1938), "Superman", *Action Comics* #1. (Siegel and Shuster 1938a) © DC Comics.

As Eisner suggests, the gestures of the comic figures are an important part of the imagery and the artist's way of conveying a story. He contends that "the human form and the language of its bodily movements become one of the essential ingredients of comic strip art. The skill with which they are employed is also a measure of the author's ability to convey his idea," and that "in comic book art, the artist must draw upon personal observation and an inventory of gestures, common and comprehensible to the reader" (Eisner 1985, pp. 100–1).

The frightened villains, Superman's heroic scenes, the humor and often times comically exaggerated gestures, and the damsel in distress are all perpetual visual motifs in the first issues of the Superman comics. The closing panel on the second page of the first issue portrays Superman holding a man above his head because he was interfering with his mission to wake up the governor to save a woman that was wrongly convicted and was to have been executed in a few hours. To get to the governor, Superman carries the man who tries to stop him, while the latter, portrayed horizontally above Superman's head, cries for help (the words 'Help! Help!' are hovering above their heads). It is a humorous panel, which shows Superman's strength while trying to avoid portraying a more serious scene (Figure 9). When Superman arrives at the governor's door, the man he had lifted a few panels earlier laughs at him while explaining that the door is locked and made of steel. Superman, undeterred, crumples the steel door, and when the other man is shocked (indicated by his open mouth and the little lines around his face), Superman standing in his Fairbanks pose tells him: "it was your idea" in a humorous, nonchalant way (Figure 5). When Superman finally gets to the governor, we see a small clock in a couple of the upper panels on the fourth page, indicating the short time left before the execution (Figure 10). This could be regarded as another 'trick' to represent the action and the time passing by in the narrative from a visual perspective (Eisner 1985, pp. 25–37; McCloud 1994, pp. 94–117). Baetens, as noted above, refers to the linear role of the text and the more spatial aspect of the illustrations. Thus, the clock bridges the two and renders the illustrations themselves more linear to fit the storyline and to guide the reader through the illustrated narrative (Baetens 2020, p. 195).

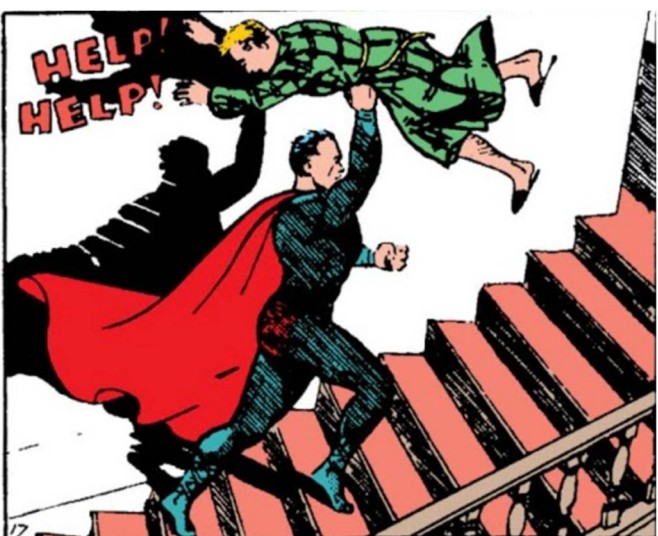

**Figure 9.** Joe Shuster, Superman Holds a Man over his Head (panel No. 17), 1938. Siegel and Shuster (June 1938), "Superman", *Action Comics* #1 (Siegel and Shuster 1938a, p. 9). © DC Comics.

Frightened villains are also seen in various panels, for example, the two villains (and a leg of a third) in the last row of the second page of the second issue (Figure 11). There, the two main figures, with a look of fright and a few lines around their heads to indicate their shock, are running from Superman after they failed to kill him, their bullets having bounced off his chest. Close-up scenes are also plentiful in those issues—showcasing a

hand about to press a button (Figure 12), a hand knocking on a door (Figure 13), and a villain shocked face when he realizes that his plan failed (Figure 14)—often indicating a moment of suspense before the action. There are many more scenes in these early issues that combine comedic elements, emotions (mainly fear), and action.

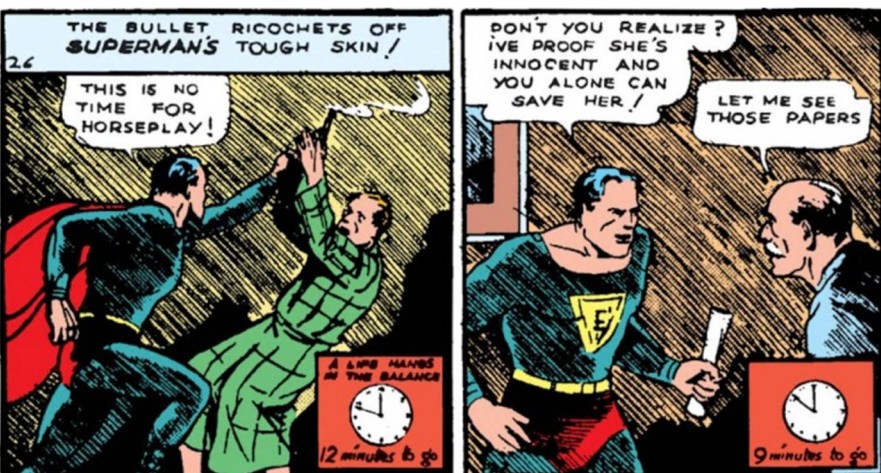

**Figure 10.** Joe Shuster, Superman Makes the Governor to Sign an Innocent Woman Acquittal (panels Nos. 26–27), 1938. Siegel and Shuster (June 1938), "Superman," *Action Comics* #1 (Siegel and Shuster 1938a, p. 11). © DC Comics.

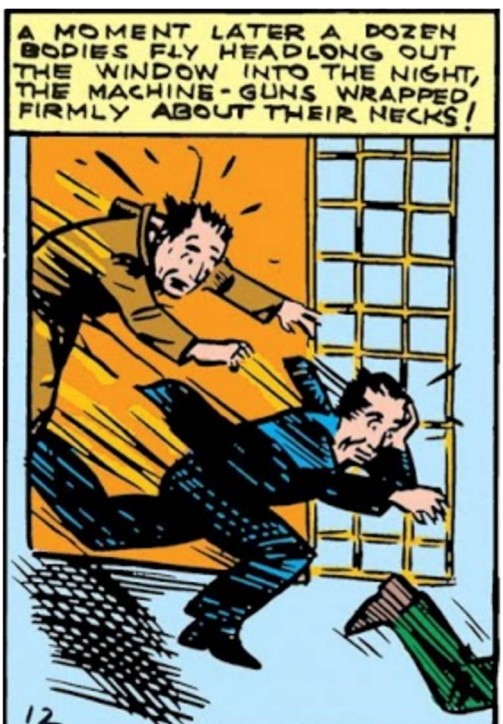

**Figure 11.** Joe Shuster, Three Armed Guards Running Away from Superman (panel No. 12), 1938. Siegel and Shuster (July 1938), "Superman", *Action Comics* #2 (Siegel and Shuster 1938b, p. 23). © DC Comics.

The humorous and comedic aspects of the comics, found in almost all of these elements (close-ups, frightened villains, Superman's approach to these villains, etc.), was partially influenced by the earlier comic strips that appeared in the Sunday newspapers, although Shuster's style was much less 'cartoony,' as he wanted to paint in a more realistic way. Thus, the humor may have made a great first impression on the readers who found *Action Comics* on the newsstands. Moreover, the emotions, the humor, and the action might also

have had a part in engaging readers with the Superman storylines; similar to going to the movies to see their favorite action character, they could find the same features in the pages of this new comic book series. To further extend the notion of the engagement of the audience as a marketing strategy, I now turn to one last visual aspect.

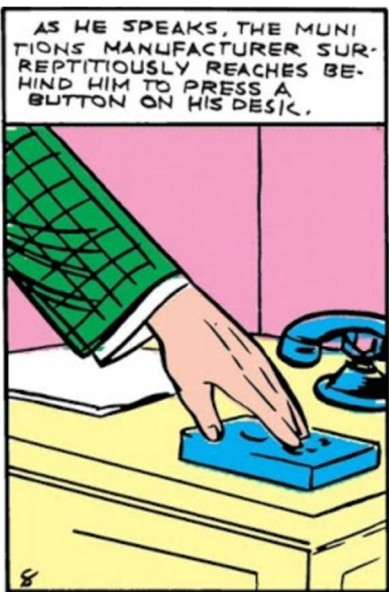

**Figure 12.** Joe Shuster, Close-Up of the Munitions Manufacturer Pressing a Button (panel No. 8), 1938. Siegel and Shuster (July 1938), "Superman", *Action Comics* #2 (Siegel and Shuster 1938b, p. 23). © DC Comics.

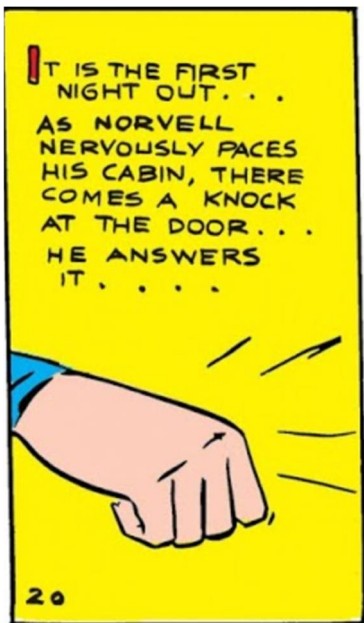

**Figure 13.** Joe Shuster, Close-Up of Superman Knocking on a Door (panel No. 20), 1938. Siegel and Shuster (July 1938), "Superman", *Action Comics* #2 (Siegel and Shuster 1938b, p. 24). © DC Comics.

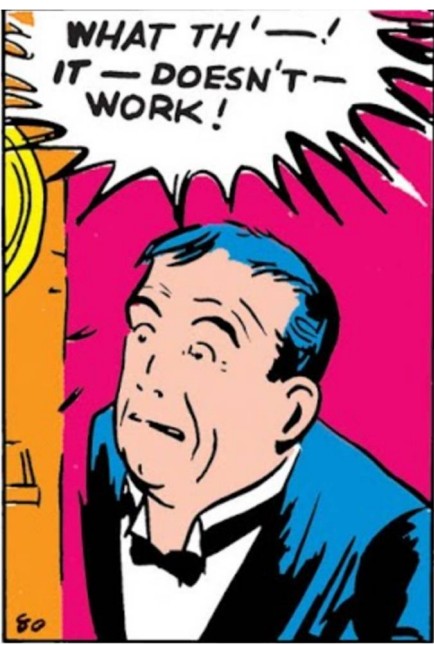

**Figure 14.** Joe Shuster, Close-Up of a Surprised Man (panel No. 80), 1938. Siegel and Shuster (August 1938), "Superman", *Action Comics* #3 (Siegel and Shuster 1938c, p. 46). © DC Comics.

## 6. Closing Panels as a Marketing Strategy

Up to this point, I have related to several visual elements that were designed to appeal to the audience in efforts to make Superman a successful figure and comics narrative. Some of these elements have been discussed in previous studies and thus are not newly discovered materials. Nevertheless, through the integration of these aspects with the visual analysis and some of the information that was conveyed by the creators themselves, I have tried to develop a new approach to reading those features in connection with the strategizing and marketing of Superman from a visual perspective. Thus far, I have demonstrated the engagement of the audience through the narrative and the design of the figures, considering a range of Superman storylines and focusing on some of the visual elements that helped engender this engagement. However, a discussion of one underrated and understudied component in the comics might well further the study of marketing-to-masses strategy: the final panel in each issue.

Each Superman storyline in the *Action Comics* series used to close with a message to the reader. The first read "And so begins the startling adventures of the most sensational strip character of all time: SUPERMAN! A physical marvel. A mental wonder. Superman is destined to reshape the destiny of the world!" and on a white square at the right side of the same panel the text said: "Only in *Action Comics* can you thrill at the daring deeds of this superb creation! Don't miss an issue!" The panel was painted in the three dominant colors of 1930s comics: blue, red, and yellow (Figure 15). Superman is portrayed on the left, breaking a chain that was wrapped around his chest, demonstrating his superstrength. This panel was clearly designed as an advertisement for Superman and combined the intriguing and alluring text, while highlighting his most prominent attribute—strength.

In the next issue, Superman once again displayed his physical attributes by showing off his biceps, while an advertisement announced the publication of the Superman comic strip in many newspapers and encouraged readers to subscribe to their local daily newspaper (Figure 16). Once again, the end panel was used to further the hype around the Superman comics. The third issue did not end with the figure of Superman but with a funny-looking boy with red hair, who reminded readers to subscribe to the "Superman of America" Society (Figure 17).[17] The fourth issue showcased a charming panel of a boy trying to acquire superstrength by lifting small objects and building his strength gradually, while a warning text about being safe was featured on the left (Figure 18). This premise, known as

"acquiring superstrength," became a recurring motif, both in the *Action Comics* series and in the succeeding *Superman* comic book series, which was first published in 1939. For example, in the sixth issue of *Action Comics*, the "acquiring superstrength" title was followed by "muscle training," which was demonstrated through illustrations picturing the way to use one's fists to acquire muscles (Figure 19); in the eighth issue, a boy was depicted trying to acquire supervision with illustrations showing the exercises (Figure 20).

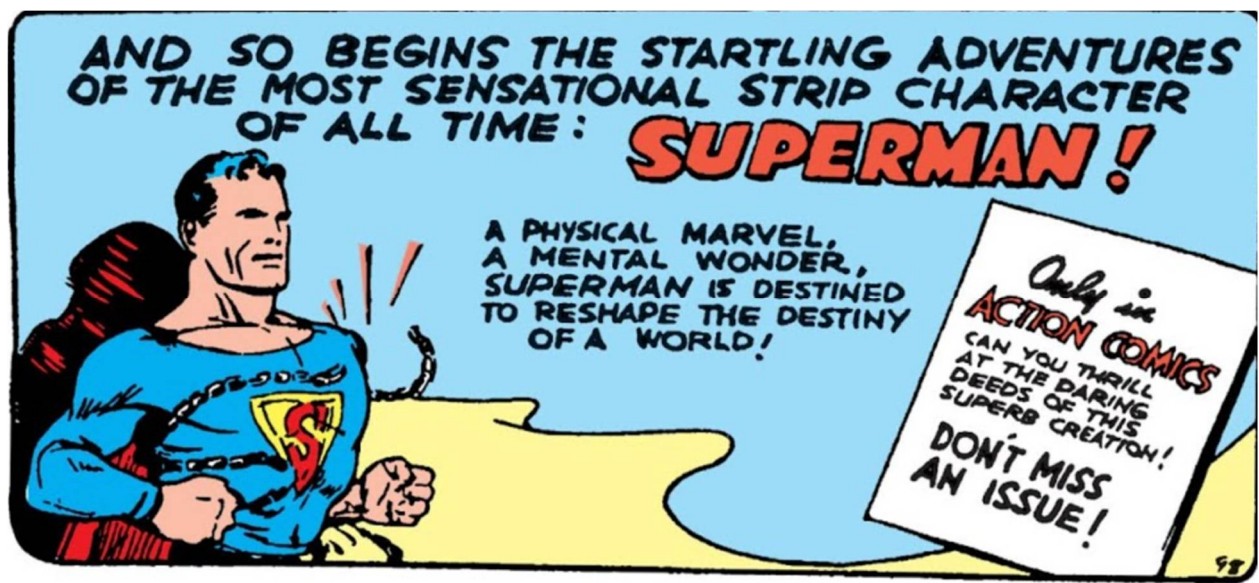

**Figure 15.** *Joe Shuster,* Closing Panel: "And So Begins the Startling Adventures of the Most Sensational Strip Character of all Time: Superman!" (panel No. 98), 1938. Siegel and Shuster (June 1938), "Superman", *Action Comics* #1 (Siegel and Shuster 1938a, p. 20). © DC Comics.

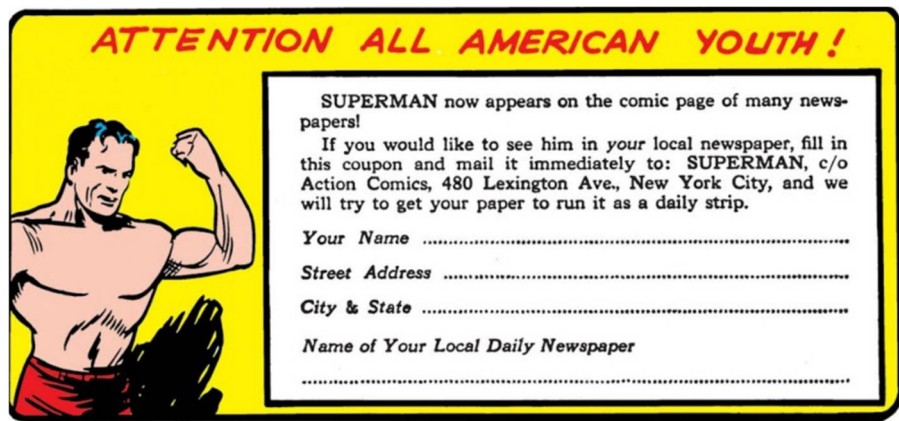

**Figure 16.** Joe Shuster, Closing Panel: "Attention all American Youth!" (panel No. 101), 1938. Siegel and Shuster (July 1938), "Superman", *Action Comics* #2 (Siegel and Shuster 1938b, p. 34). © DC Comics.

These panels were a constant feature in most of those issues and were clearly designed to engage the audience with the storylines and the Superman figure, encouraging them to buy the next issues and also access other platforms to consume more Superman content. To achieve involvement, these panels did not merely advertise the content of the comics but included relevant imagery: Superman's demonstrating his superhuman powers, tips on how to acquire superstrength, illustrating it with a young boy to whom the audience could relate, etc. It is important to note that even though these panels closed each Superman chapter in *Action Comics*, there were no similar panels in any of the other comics narratives in these issues. As noted, these end panels have been largely overlooked by scholars in the field, perhaps because it was thought that they did not have any narrative significance.

Even so, I suggest that these end panels played a part in the consumption of these comics storylines and the rise in Superman's popularity by encouraging readers to engage and even become part of the narrative.

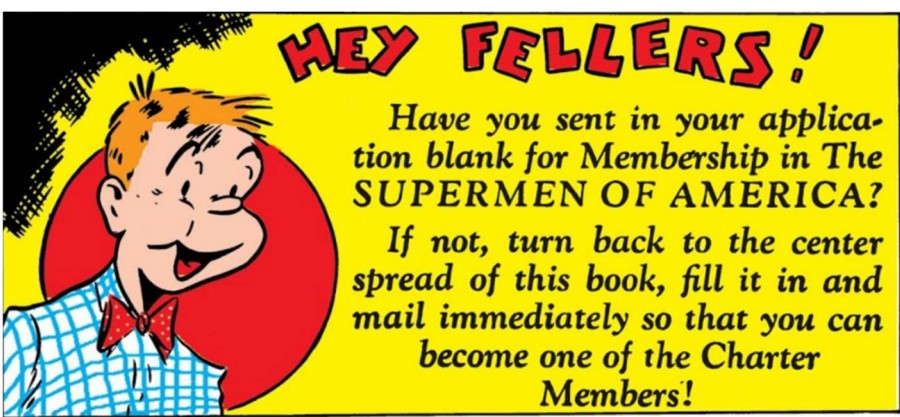

**Figure 17.** Joe Shuster, Closing Panel: "Hey Fellers!" (panel No. 96), 1938. Siegel and Shuster (August 1938), "Superman", *Action Comics* #3 (Siegel and Shuster 1938c, p. 48). © DC Comics.

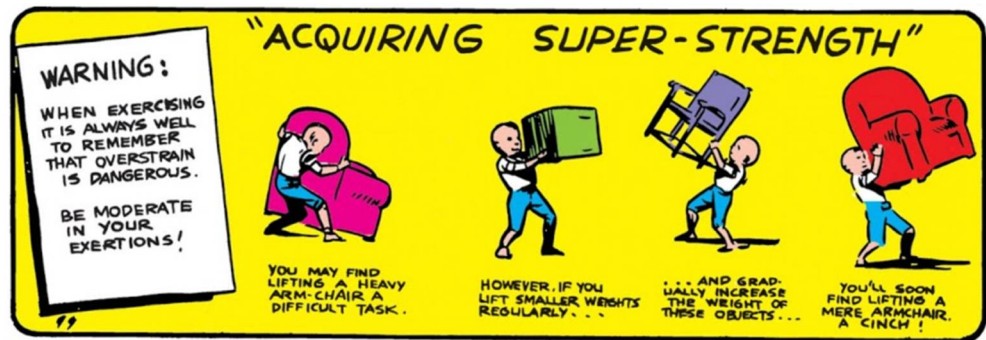

**Figure 18.** Joe Shuster, Closing Panel: "Acquiring Super-Strength" (panel No. 99), 1938. Siegel and Shuster (September 1938), "Superman", *Action Comics* #6 (Siegel and Shuster 1938d, p. 62). © DC Comics.

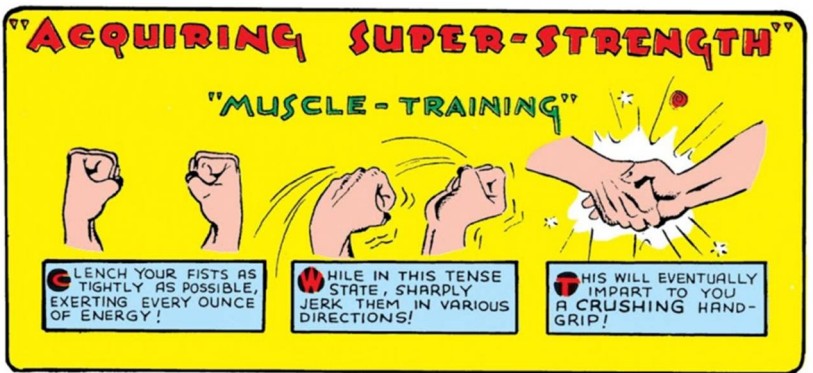

**Figure 19.** Joe Shuster, Closing Panel: "Acquiring Super-Strength" (panel No. 100), 1938. Siegel and Shuster (November 1938), "Superman", *Action Comics* #6 (Siegel and Shuster 1938d, p. 86). © DC Comics.

Although these panels were not a primary component in the comic books of the 1930s, as, for example, one of the first and most famous comic books series of that time, *Famous Funnies*,[18] did not end its narratives in that way, a similar approach can be seen, with slight changes, in the next superhero comics that Detective Comics published. Batman, which first appeared in its Issue No. 27 in 1939. In the Batman Comics there was a closing panel that served more a "to be continued" kind of function, encouraging readers to stay tuned for the next issues. However, it did not elicit the kind of audience engagement that

was found in the Superman closing panels, which not only invited the readers to stay tuned, but also were designed to bring them into the narrative—for example, to try and be superstrong like Superman.

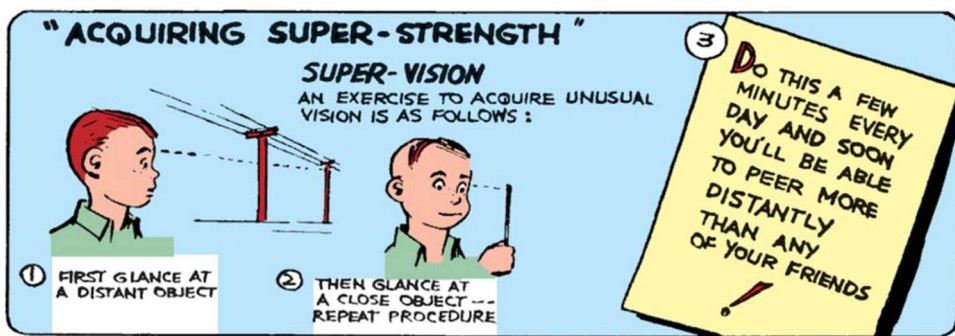

**Figure 20.** Joe Shuster, Closing Panel: "Acquiring Super-Strength," 1938. Siegel and Shuster (January 1939), "Superman", *Action Comics* #8 (Siegel and Shuster 1939, p. 114). © DC Comics.

Thus, these panels were unique hybrids of narrative and advertisement, on the one hand, encouraging the audience to continue to read the comics, building a sort of hype around the new superhero and, on the other hand, urging the reader to get involved in the narrative itself. Many of these comics' young readers might have wanted to look and become like Superman with his superhuman abilities. These closing panels, which were there from the first issues, tended to make these aspirations appear possible to some degree, probably helped with the promotion of the Superman narrative, and had a role in its rise to popularity. One might assume that if these panels did not work, they would not have been continued in subsequent issues and in the Superman series that followed. Thus, these panels, which combined text, illustrations, and advertisements, offer an interesting case study.

## 7. Conclusions

Vincent Sullivan, the editor of *Action Comics* who worked with Siegel and Shuster in connection with the early issues, once mentioned that the reason Detective Comics agreed to publish the Superman narrative after the creators submitted their comic strip was that "It looked good. It was different and there was a lot of action. This is what the kids wanted" (Harvey 1996, p. 18).

Returning to my earlier discussion about the different approaches that could be relevant when talking about artistic marketing strategies, one could say that what follows can be viewed in the light of a patron-artist approach. Sullivan indicated that he accepted the strip both because he thought it looked good and because he saw it as, perhaps, a financial opportunity in selling it to a young American audience. This leads to the second approach, that is, the creators-audience aspect, since the patron, if one can call a publishing house a patron, also addressed the audiences as consumers of the product/art.

To understand the importance of visuals vis à vis the narratives in the first few issues of the Superman comic books, I considered several visual elements that helped to promote them: the design of Superman and other archetypal figures, the illustrated narrative, and the closing panels. The figure of Superman was conceived to appeal to the audience through several contemporary fashionable trends: manly Hollywood actors, a silent-movies protagonist, several pulp fiction heroes, and famous men from the bodybuilding world, known from fitness magazines among other media. All of these had a part in shaping the ideal man, who, on the one hand, was "out of reach" because of his ideality and, on the other hand, was a figure the audience wanted to be and to look like. In the illustrated narrative, I focused on three prominent components: action, humor, and emotions, all of which contributed to the audiences' engagement with the narrative and revealed many of the characteristics found in the movies of that era. The closing panels, a recurring feature in the Superman narratives in *Action Comics*, which has been largely overlooked so far,

further contributed to readers' engagement by combining advertisements, narrative, and visual appeal. These panels occasionally presented the figure of Superman demonstrating his superhuman powers accompanied by text that encouraged the audience to buy the subsequent issues, while also offering them other platforms by which they could consume other Superman narratives and occasionally presented the "How to acquire superstrength" caption, encouraging the young audience to be more like Superman, which, in some way, enabled entry into the fantastical realm of the superhero's world.

The fourth issue, published in September 1938, marked the real beginning of Superman's rise in popularity, and from there on *Action Comics* started to sell widely, with a circulation by 1939 of 500,000 monthly copies, and by 1940 more than a million copies were sold each month (ibid., p. 19). Shortly after the first few issues, Shuster had to hire a crew of illustrators to help him keep up with the *Action Comics* series, the *Superman* series, and the *Superman* comic strips that were published in numerous daily newspapers (Gordon 2017, p. 98). It seems that the young creators' dream had come true, with Superman read and known widely throughout the United States.[19]

There is no doubt that even though Siegel was the leader in the partnership, writing the storylines and the original creator of the Superman character, Shuster had a major part in their triumph (Gordon 2017, p. 98). After the initial success of the Superman comics, other superheroes started to appear, and a new genre evolved, which is a feature in pop culture until this day, with many superhero movies and TV shows debuting each year. Superhero comic books are still being published, including the *Action Comics* series that still features Superman, and although the movies have somewhat replaced comic books as the major outlet for these storylines, the former still enjoy a large readership.[20]

Superman can be regarded as the source of a new iconography—that of the comics superhero. Although recent decades have seen an increase in diversity among superheroes and superheroines, the strong, muscular, sleek black-haired hero still dominates the medium. The years from 1938 to the 1950s, known as the Golden Age of Superheroes, saw the emergence of many superheroes that copied Superman's appearance, for example, Wonder Man, who was introduced in 1939, at which time his creators were sued for copyright violations. Captain Marvel (known today as Shazam), Captain America, and many others appeared subsequently.[21] Even the actors who played Superman over the years, from Christopher Reeves to Henry Cavill seemed to fit the same image, which matches the iconography, which moved from comic books to the movies and back.

It is clear that by making use of many elements that were familiar and appealing to the series audiences, the visual design and illustrations in the first few issues of *Action Comics'* Superman storyline played a large part in the rise and success of the Superman character, both as a figure and as a comic book narrative. All of the relevant aspects included the audience in the comics themselves, exhorting them to look at Superman and aspire to be like him, to identify with Clark Kent, and to enjoy the marvelous action and humorous events that unfolded.

Throughout this paper, I have used art history methodologies and vocabulary to describe and analyze the illustrated aspect of the comics, such as the concepts of "patron" and "iconography." Thus, one question that might come up from such discussion is whether comic books, and the superhero genre, can be considered art? I suggested, for example, that the publishing company and the audiences could be considered "modern-day patrons," and that the illustrations may have been a prominent component in marketing the early comic book issues. Many scholars have tried to answer similar questions and the one most asked often in comics studies is whether the subject of comics can be regarded as a legitimate field of research? (Duncan and Smith 2012; Stein 2018, pp. 259–71; Hatfield and Beaty 2020). The answer can be found in the fact that in recent years, the disciplines of art and art history have been largely expanded to include other aspects of visual culture, including mass media and pop culture, both of which embrace comics in general and comic books in particular.[22] Thus, when addressing comic books in general, as well as those early comics issues that engendered the superhero genre, I suggest that one can also look at them

through the prism of art history. The illustrations, accompanying or being accompanied by text in a symbiotic way, more so than the book illustrations in the Middle Ages and the Renaissance, encourage readers as well as scholars to ask new questions, while leaning on earlier methodologies to do so. Based on the recent discourse, I expect that more art historians will look at the comics medium as an art form and ask similar questions in order to develop this unique discipline.

**Funding:** This research was funded by the Rotenstreich scholarship for outstanding doctoral students in the humanities in Israel.

**Acknowledgments:** I am Grateful to Daniel M. Unger for his valuable insights and support. I am thankful to Nirit Ben-Aryeh Debby, Inbal Ben-Asher Gitler, Priel Cohanim, Adi Hamer Yacobi, Ruth Noyes Sargent, Simon Abrahams, Lydia Goodson, and Ragnhild M. Bø for their valuable insights and generous help. I also thank the two anonymous readers for their thorough reading and valuable suggestions.

**Conflicts of Interest:** The author declares no conflict of interest.

## Notes

1    For further reading on the superhero genre in general, see Reynolds (1992); Coogan (2006); Chambliss and Svitavsky (2013, pp. 6–27); Singer (2020, pp. 213–26).

2    On the pulp fiction hero Tarzan, see De Silva (2004).

3    It is important to note that throughout the years, Shuster and Siegel interviewed for various media sources to tell their story of the first years of Superman's publication. They were involved in legal disputes with DC comics throughout the years, and they often disparaged the industry process. Many of the legal files from these disputes were uploaded to SCRIBD by Jeff Trexler: https://www.scribd.com/user/3105491/Jeff-Trexler/uploads (accessed on 27 August 2021).

4    Specifically for the connection between superheroes and mass culture, see Eaton (2013, pp. 28–39).

5    Apart from the 1986 interview in *Nemo: The Classic Comics Library* (Andrae et al. 1986, pp. 6–19), much of the research conveys the early years of the Superman phenomenon from a historical-biographical point of view. See, e.g., Harvey (1996, pp. 16–49); Daniels (1998); Andrae and Gordon (2010); Lund (2016); and Gordon (2017).

6    For the complete "The Reign of the Superman" series, see Andrae et al. (1986, pp. 20–47) and Sesselego and Livi (2020, pp. 29–46).

7    All figures and information of the first comics issues were taken from Volume 1 of *The Golden Age Superman*, which includes *Action Comics* Nos.1–19 (June 1938–December 1939), Superman comics Nos. 1–3 (July 1939–Winter 1939), and other Superman storylines. All further references to this volume will be indicated with the initials *GAS* with page numbers and the complete reference information regarding the particular issue. Woodard and Santos (2016); Siegel and Shuster (June 1938), "Superman", *Action Comics* No. 1 (Siegel and Shuster 1938a, p. 8).

8    Siegel and Shuster (Siegel and Shuster 1938a, pp. 8–20; ibid., pp. 21–34; ibid., pp. 35–48).

9    Two issues were published, one in 1939 and one in 1940: Jerry Siegel and Joe Shuster, "Superman at the World's Fair", issues #1 and #2, *New York's World Fair Comics*, April 1939 and July 1940. The second issue also mentions "Superman Day".

10    Siegel and Shuster *GAS*, p. 8.

11    For a full analysis of ancient sources on Herakles, see Stafford (2011).

12    For example, on the cover of the first *Action Comics* issue (presented further on in this paper), Superman is described holding a car above his head. This can be associated with the mythological figure Atlas, who was often described holding the globe above his head (even though he was meant to hold the heavens, and not earth, for eternity). Another association was with Hercules battling various mythological monsters. Antonio del Pollaiolo's 1475 painting featuring Hercules fighting the Hydra can be can be compared to the position in which Superman in portrayed on the cover of the first Action Comics issue. Pollaiolo's Hercules is imaged in a similar gesture, holding his club above his head instead of a car. See Antonio del Pollaiolo, *Hercules and the Hydra*, 1475, Firenze: Le Galerrie degli Uffizi. That is not to say that Shuster took direct inspiration from this Renaissance painting, but rather that Superman embodies the charactaristic of the heroic figure, both conceptually and visually.

13    A relatively recent reference to the red trunks can be found in *Action Comics* No. 697, where Superman's son finds a picture of his old costume on his father's phone and calls them "the undies on the outside." See *Action Comics* No. 697, 2017.

14    On masculinity and actors in 1930s Hollywood, see Mellen (1977, pp. 96–138); Neibaur (1989, pp. 1–15, 114–18); LaSalle (2002, pp. 1–16, 60–76).

15    Siegel and Shuster *GAS*, p. 10.

16    For a comprehensive study on the history, development, and style of comics strip, see Gordon (1998).

17    On the *Superman Society of America*, see Daniels (1998, pp. 46–47).

18    On the *Famous Funnies* comic books, see Hatfield (2020, pp. 27–28).
19    On Superman merchandizing through the years, see Daniels (1998, pp. 130–84).
20    On contemporary readership and communities of fans, see, e.g., Beaty (2016, pp. 318–25).
21    On the subject of the ages of superheroes, see Oropeza (2005), pp. 10–18.
22    See, e.g., Mirzoeff (1999); Danesi (2019).

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
