# Peer review of "Visualizing Superman: Artistic Strategizing in Early Representations of the Archetypal Man in Comic Books"

_arts, 2021_

Round 1

Reviewer 1 Report

Dear author:

This is a fine article. It's good to see an art historical perspective on superhero comics. Perhaps this perspective could be strengthened a bit (the term "iconography" comes up but is used in a rather conventional arther than specifically art historical way, it seems to me).

I appreciate that you found and use the 1983 interview from Nemo, and I feel like it's puts to good use. Perhaps a caveat about the retroactive nature of Siegel and Shuster comments would be good, though. After all, they have a vested interest in telling this particular story about Superman's conception, and they tend to downplay the industry processes that Lund and also Gordon discuss at length.

L. 116: I wouldn't necessarily say that "little has been written about its visual aspect"; there is, I believe, quite a bit scholarship that talks about superhero imagery, though not necessarily from an art history perspective or through the lens of style.

L. 346: Is there a date for figure 6 that could be added?

L. 465: Superman lifting the car on the cover of Action Comics #1 reminds me of classical depictions of Atlas carrying the world.

L. 467 and passing: I'm not sure whether the scenes/images that are discussed as humorous are always humorous. In terms of the cover, comically exaggerated would sound more appropriate to me.

None of these suggestions and comments take away from the overall validity and originality of this fine essay.

Author Response

I would like to thank the two anonymous reviewers for their helpful suggestions and insights and their kind words that encouraged me to further improve my paper. I enclose here the list of my corrections, following the reviewers’ comments. All corrections are written in red in the revised text.

Reviewer 1

I would like to thank the first reviewer for his/her wonderful and helpful insights, and for kind words at the beginning and end of the review.

  • This is a fine article. It's good to see an art historical perspective on superhero comics. Perhaps this perspective could be strengthened a bit (the term "iconography" comes up but is used in a rather conventional rather than specifically art historical way, it seems to me).
    I added a paragraph dedicated to terminology and how I, from an art history point of view, approach the analysis of the Superman comics in question. See lines 127-143. This paragraph is followed by another that addresses a comment of the second reviewer (lines 144-155).
  • I appreciate that you found and use the 1983 interview from Nemo, and I feel like it's puts to good use. Perhaps a caveat about the retroactive nature of Siegel and Shuster comments would be good, though. After all, they have a vested interest in telling this particular story about Superman's conception, and they tend to downplay the industry processes that Lund and also Gordon discuss at length.
    I added a footnote addressing this issue. See note 3.
  • 116: I wouldn't necessarily say that "little has been written about its visual aspect"; there is, I believe, quite a bit scholarship that talks about superhero imagery, though not necessarily from an art history perspective or through the lens of style
    I deleted this statement and kept the statement that: “comic book art has not yet been accorded much recognition among art historians.” See lines 116-117.
  • 346: Is there a date for figure 6 that could be added?
    Unfortunately, no source that I found indicates the date for the figure. However, it might be a limited edition that Shuster created and signed for fans in the 1970s. Although this is a later image than the period in question in this paper, I found it useful to present the Superman figure that was established throughout the years. Thus, I added “Probably 1970s” to the image. See page 16.
  • 465: Superman lifting the car on the cover of Action Comics #1 reminds me of classical depictions of Atlas carrying the world.
    I thank the reviewer for this wonderful insight. Although I don’t have the space to address it fully, I added a footnote in this subject. This could be a wonderful idea for a future paper. See note 12.
  • 467 and passing: I'm not sure whether the scenes/images that are discussed as humorous are always humorous. In terms of the cover, comically exaggerated would sound more appropriate to me.
    I adjusted some of the sentences where I find this term very useful and more correct – as the reviewer suggests. See for example lines 512, 526-7, 563, 573. [I used the word “comedic” instead of “comic” in some cases, to differentiate from the comics medium].

Reviewer 2 Report

Tom Andrae's name is repeatedly misspelled. The manuscript probably needs a closer proofreading.

Key terms, such as "visual design" and "illustrated narrative" are not clearly defined early on, and there seems to be some slight shifting of terms in the course of the article. Does "appealing narrative" (lines 45-46) overlap with "illustrated narrative," or does appealing narrative only refer to the appeal of the characters - Superman as an aspirational figure and Clark as a character with whom the readers can identify - while illustrated narrative only refers to the art style used to depict action, humor, and emotion? I read it that way, but I am not confident in my interpretation because the article did not always provide a clear and precise use of terminology.

Another example of shifting terms:  On line 36 you ask "did the visual design account for a major part in that phenomenal success?", but when you answer that question on lines 701-703 you use the terms "illustrations". I do not consider illustrations and visual design to be equivalent terms.

On lines 551-552 you acknowledge that "Some of these elements have been discussed in previous studies and thus are not newly discovered materials." You do a good job of citing sources for additional information, and it is important to cover that previously covered ground because most of your reader will not know most of that history or have been exposed to most of that theory. However, you do a bit of a disservice to your reader because you not not provide some information that is particularly relevant to your article. For instance, you mention the Superman balloon in Macy's Thanksgiving Day Parade as an indicate of the character's popularity, but you do not mention the Superman merchandise in the Macy's store, Superman Day at the World's Fair (July 1940), or the Superman Adventure at Macy's (November 1940). A pretty big promotional push for a character that had only been around a couple of years.

Also, you could have gone a bit deeper when talking about the sources of inspiration for the Superman character. I'm don't know if Joe Shuster ever saw this magazine, but in the early 1930s there was a physical culture magazine titled Superman (see The Power of Comics: History, Form and Culture, 2nd edition page 195). The fit, mostly nude men on the covers were often in heroic poses. One of the most important images in those early issues was Superman leaping over tall buildings. That leaping ability was borrowed directly from John Carter of Mars, and even the explanation provided in the comics, the slighter gravity pull of Earth, came from the John Carter books. The most glaring omission in the sources section is no mention of Phillip Wylie's 1930 novel Gladiator. Some of the captions and dialogue in early issues of Action and Superman are extremely close to phrases used in Wylie's novel.

A good analysis of how Shuster combined a "matinee idol" heroic figure in action with some of the visual conventions of humor comic strips. I don't think there has been enough written about the techniques Shuster used to convey humor and emotion.

Author Response

I would like to thank the two anonymous reviewers for their helpful suggestions and insights and their kind words that encouraged me to further improve my paper. I enclose here the list of my corrections, following the reviewers’ comments. All corrections are written in red in the revised text.

Reviewer 2

I would like to thank the second reviewer for his/here helpful insights and drawing my attention to several points I didn’t include in the previous version of the paper.

  • Tom Andrae's name is repeatedly misspelled. The manuscript probably needs a closer proofreading.
    I corrected the spelling of Andrae’s name and proofread my paper.
  • Key terms, such as "visual design" and "illustrated narrative" are not clearly defined early on, and there seems to be some slight shifting of terms in the course of the article. Does "appealing narrative" (lines 45-46) overlap with "illustrated narrative," or does appealing narrative only refer to the appeal of the characters - Superman as an aspirational figure and Clark as a character with whom the readers can identify - while illustrated narrative only refers to the art style used to depict action, humor, and emotion? I read it that way, but I am not confident in my interpretation because the article did not always provide a clear and precise use of terminology.
    I added a paragraph addressing this issue. See lines 144-155. This paragraph is preceded by another paragraph that addresses a comment by the first reviewer (lines 127-143).
  • Another example of shifting terms:  On line 36 you ask "did the visual design account for a major part in that phenomenal success?", but when you answer that question on lines 701-703 you use the terms "illustrations". I do not consider illustrations and visual design to be equivalent terms.
    I corrected the sentence to “the visual design and illustrations.” See line 741.
  • On lines 551-552 you acknowledge that "Some of these elements have been discussed in previous studies and thus are not newly discovered materials." You do a good job of citing sources for additional information, and it is important to cover that previously covered ground because most of your reader will not know most of that history or have been exposed to most of that theory. However, you do a bit of a disservice to your reader because you not not provide some information that is particularly relevant to your article. For instance, you mention the Superman balloon in Macy's Thanksgiving Day Parade as an indicate of the character's popularity, but you do not mention the Superman merchandise in the Macy's store, Superman Day at the World's Fair (July 1940), or the Superman Adventure at Macy's (November 1940). A pretty big promotional push for a character that had only been around a couple of years.
    This is an interesting comment, and I added some information about it. Since it is not the main subject of the paper, I couldn’t elaborate extensively, but I agree with the reviewer that these are important details. I included the information in lines 267-270 + note 9.
  • Also, you could have gone a bit deeper when talking about the sources of inspiration for the Superman character. I'm don't know if Joe Shuster ever saw this magazine, but in the early 1930s there was a physical culture magazine titled Superman(see The Power of Comics: History, Form and Culture, 2nd edition page 195). The fit, mostly nude men on the covers were often in heroic poses. One of the most important images in those early issues was Superman leaping over tall buildings. That leaping ability was borrowed directly from John Carter of Mars, and even the explanation provided in the comics, the slighter gravity pull of Earth, came from the John Carter books. The most glaring omission in the sources section is no mention of Phillip Wylie's 1930 novel Gladiator. Some of the captions and dialogue in early issues of Action and Superman are extremely close to phrases used in Wylie's novel.
    I agree that further information should be presented when discussing Shuster and Siegel’s inspirations for designing Superman. Since much of the information is included in various studies, I only presented it briefly. I mentioned in the previous version the connection to some of the pulp magazines heroes and included John Carter of Mars among them. I thank the reviewer for the reference to the publication The Power of Comics: History, Form and Culture, which is now included in this paper and bibliography. I mention in this revised version the physical culture magazine entitled Superman, and the connection to the 1930 Gladiator See lines 399-403 and 412-416.
